# Reply to Wolf et al.: Why Trap-Neuter-Return (TNR) Is Not an Ethical Solution for Stray Cat Management

**DOI:** 10.3390/ani10091525

**Published:** 2020-08-28

**Authors:** John L. Read, Chris R. Dickman, Wayne S. J. Boardman, Christopher A. Lepczyk

**Affiliations:** 1Earth and Environmental Sciences, University of Adelaide, Adelaide, SA 5000, Australia; 2School of Life and Environmental Sciences, University of Sydney, Sydney, NSW 2006, Australia; chris.dickman@sydney.edu.au; 3School of Animal and Veterinary Sciences, University of Adelaide, Adelaide, SA 5000, Australia; wayne.boardman@adelaide.edu.au; 4School of Forestry and Wildlife Sciences, Auburn University, Auburn, AL 36849, USA; cal0044@auburn.edu

**Keywords:** animal welfare, cat, euthanasia, infectious disease, shelters, trap-neuter-return, TNR, urban stray, wildlife

## Abstract

**Simple Summary:**

We provide a rebuttal to Wolf et al. (2019), outlining biological, ethical, and economic flaws in their argument that Trap-Neuter-Return (TNR) is an ethical solution for stray cat management. We contend that suppression of supplementary feeding of stray cats accompanied by proactive adoption or prompt euthanasia is more effective, humane, and economical.

**Abstract:**

We critique the recent article by Wolf et al. (2019) that claims scientific merit for reducing the number of stray cats in Australia through Trap-Neuter-Return (TNR) programs, and then we provide an inventory of biological, welfare, and economic reasons why TNR is less successful than adoption and euthanasia for managing unowned cats. Like Crawford et al. (2019) and multiple other comprehensive and unbiased Australian and international scientific reviews, we refute the idea that returning neutered unowned cats to stray populations has any valid role in responsible, ethical, affordable, and effective cat management, or in wildlife conservation. The main purported objective of TNR proponents along with animal welfare, human health, and wildlife advocacy stakeholders is to reduce the number of unhomed cats. We contend that cessation of provisioning unowned cats with food is the most effective approach to achieve this objective. We also present evidence from the Brisbane City Council that informed cat management policy, advocacy, and laws, backed up by responsible rehoming or prompt ethical euthanasia, are together effective at reducing the stray cat problem.

## 1. Introduction

Optimal management of unowned domestic cats (*Felis catus*) is important for improving animal welfare, wildlife conservation, human health, and minimising social conflict issues in Australia [1] and around the world [2,3,4]. Whilst the imperative to control free-ranging feral cats to protect wildlife [1] along with reducing disease burdens to humans [4,5,6], livestock and wildlife [7,8] is broadly supported [9], socio-political policies for managing stray cats living in urban or peri-urban environments with anthropogenic food subsidies are more nuanced and controversial [10,11].

Two recent publications in *Animals* present dramatically different perspectives on the efficacy and justification for an Australian trial of releasing neutered cats to reduce unowned cat numbers. “A Case of Letting the Cat out of the Bag—Why Trap-Neuter-Return Is Not an Ethical Solution for Stray Cat (*Felis catus*) Management” by Crawford et al. [12] refutes advocacy for TNR trials in Australia by citing a number of flaws in this approach from international and Australian data. In contrast, “Reply to Crawford et al.: Why Trap-Neuter-Return (TNR) Is an Ethical Solution for Stray Cat Management” by Wolf et al. [13] claims Crawford et al. present “a position of opposition to trap-neuter-return (TNR) that conflicts with the available scientific evidence and wrongly concludes that trialling the effectiveness of TNR should not be attempted in Australia.”

In evaluating both papers, we note a number of issues that require clarification and attention. First, we provide more plausible and defendable explanations of key elements of the Wolf et al. [13] rebuttal paper. We do not comment on their specific criticisms of the Crawford et al. [12] paper, which have been dealt with in an independent response [14], rather we challenge general statements made by Wolf et al. about the efficacy and ethics of TNR. Second, we respond to the assertion by Wolf et al. that their response “provides contextual background for this important animal and human issue.” We address this contextual clarification by outlining underlying biological, sociological, and economic flaws in TNR logic that account for its inefficient and unethical outcomes. Finally, after synthesising scientific, sociological, and economic evidence and in the interests of pursuing a solution that reduces the ethical, biological, health, and social challenges of unowned cats, we propose that Australia continues along the path of promoting responsible pet ownership, while reducing food subsidies for unowned cats. We use a case study example to make this last point.

## 2. Critique of Wolf et al.

Five statements (S) in the Wolf et al. [13] rebuttal article are listed *in italics* and addressed in our accompanying response (R):1SWolf et al. [13] argue that “*all key stakeholders, including groups that support or oppose TNR, seek the common goal of sustainably reducing the number of unowned urban stray cats in the long term*.”1RWe concur that reduction in stray cat numbers to very low levels that are sustained indefinitely should be the primary objective of unowned cat management and the main criterion upon which management success is judged. However, reduction in euthanasia rates and lessening the emotional stress of cat management personnel, whilst important considerations, are not key success indicators of this primary objective, as repeatedly touted in Wolf et al. [13]. Neither euthanasia rates nor the stress borne by cat management personnel necessarily bear a strong relationship to outdoor cat numbers or management and policy approaches to reduce the number of unowned urban cats.2SWolf et al. [13] argue that “*strong scientific evidence shows that when implemented with sufficient intensity and combined with adoption efforts (as is common practice), TNR can significantly reduce the number of unowned stray cats in urban areas*.”2RAvailable scientific evidence, including two recent independent and comprehensive Australian reviews [1,15] not referenced by either Crawford et al. [12] or Wolf et al. [13], support the weight of peer reviewed evidence [16,17] that TNR does not effectively reduce the number of unowned cats. Adoption of stray cats, which should occur independently and irrespective of TNR, is the principal tool for reducing stray cat numbers in clowders that are also “managed” by TNR advocates. Neutering and adoption rates need to be individually or collectively maintained at 70% to 75% to halt population growth [18,19]. Although the assertion by Wolf et al. [13] that TNR can, rather than does, significantly reduce the number of unowned cats is recognised as a disclaimer, this point demands clarification because purported, yet very rarely achieved or reported, reduction in unowned cat numbers is the main driver for the Best Friends sponsored advocacy for TNR trials in Australia. No studies have demonstrated that TNR reduces a cat population to zero, even in “colonies”, within the lifespan of a cat.3SWolf et al. [13] argue that “*current cat management methods across Australia, which have been undertaken routinely for many years, simply have not decreased the number of unowned urban stray cats in Australia. For example, despite killing 118,000 out of the 196,000 cats impounded by local government from cities and towns across New South Wales over an 8-year period (2008–2009 to 2015–16; 60% euthanasia rate), there was virtually no change in cat intake (25,000 in 2008–2009 vs. 24,000 in 2015–2016)*.”3RWe do not have evidence to disagree with the assertion of Wolf et al. that current management in most jurisdictions has failed to significantly reduce the number of unowned cats and we agree that policy, and particularly proactive management actions, require an informed overhaul. However, we disagree that cat intake to shelters necessarily provides a good measure of stray populations and strongly refute the idea that euthanasia of many cats surrendered to Australian shelters is the cause of contemporary cat management failings. With the exception of the Brisbane City Council (details provided below), most jurisdictions typically employ reactive, ad hoc trapping of a small percentage of problem cats, leaving the vast majority of unowned urban cats entirely unmanaged and not exposed to the “*high level of killing*” emotively described in the paper. The key failure of contemporary cat management in Australia is that the vast majority of unowned urban cats are unmanaged, which is not assessable by the percentage of cats impounded by local government that are euthanased.4SWolf et al. [13] state that “*We agree with Crawford* et al. *that “capturing, transporting, neutering, vaccinating, worming and medicating are stressful procedures even for well-socialized pet cats, let alone for stray cats unsocialized/partially socialized to human contact*.”4RWe agree with Crawford et al. [12], Wolf et al. [13], and the RSPCA that protracted medical and management intervention is stressful to cats, especially unowned and unsocialised cats. Like People for the ethical Treatment of Animals (PeTA), we consider that the stress to cats imposed by TNR is one of the reasons that TNR is unethical. Instead, we propose that the onus should be placed on cat owners to identify and register their cats through microchipping, as a minimum, and to maintain them on their premises or under control (just like dog owners), so that the management of unowned cats can occur far more expediently, with less stress to cats and cat-care professionals.5SWolf et al. [13] state that “*The proposed ‘targeted adoption’ program, which involves “rehabilitation” of unsocialized cats, would very likely result in much greater costs (as well as overcrowded shelters and an increase in the number of cats killed because of space limitations). The authors acknowledge the “clear need for economic research on the relative costs and effectiveness of different proposed strategies for reducing numbers of stray cats in Australian cities” even as they appear to accept, uncritically, the results of a dubious model based on a “single super colony [of 30,000 cats]”. We too welcome additional economic analysis; however, the available evidence strongly suggests that the costs associated with expanded adoption efforts will likely exceed the cost of TNR*.”5RWe appreciate that Wolf et al. agree on the need for further economic research. However, we would note that the authors provide no data or literature to support their claim that the targeted adoption program or expanded adoption programs would cost more that TNR. While an adoption program is not free, full cost accounting in a comparative manner needs to be done in order to demonstrate what these approaches actually cost. We would further note that the “dubious” model [20] cited by Wolf et al. did show full cost accounting and was published in a well-regarded peer reviewed journal. Finally, simply providing accounting for costs is incomplete as the economic analyses need to be contextualized with respect to what they achieve in terms of reducing the number of stray cats. For instance, an adoption program may have a greater total cost, but be less expensive on a per capita basis, and therefore more efficient.

## 3. Biological Flaws in Applicability of TNR to Stray Cat Management

Although the release of neutered animals can successfully limit populations of short-lived insects that typically mate only once [21], and reduced populations of non-pest animals can theoretically be achieved through fertility control (e.g., [22,23]), three characteristics of cat biology or behaviour are inconsistent with TNR providing a viable technique for sustainably eliminating stray cat populations.

### 3.1. Unachievable Trapping Rates

The sterilization of feral cat populations requires that an exceptionally high rate of individuals are spayed/neutered, which makes this tool essentially useless if these rates cannot be achieved. Even in a fully closed population (i.e., with no immigration or abandonment), maintenance of sterilization rates exceeding 70% [19,24] is typically unachievable, especially in large clowders that include cats with an inherent or learned aversion to being trapped or when abandonment of cats is prevalent. Lethal control, which could include techniques that either do not require cats to be captured or require one capture at most, do not face the same limitations as TNR, which requires multiple captures to ensure effective immunisation.

### 3.2. Lack of Mate Defence

TNR proponents claim that neutered cats reduce the reproductive rates of intact cats (both resident and immigrants) by restricting access to queens on heat. However, since ovulation in cats is triggered by multiple copulations [25], queens in oestrus actively seek out multiple males. Furthermore, even intact tom cats that mate with a queen seldom defend her from subsequent matings, as multiple-siring of most litters of free-ranging cats attests [2]. The biologically incorrect terminology “cat colony”, which implies social cohesion and structure that could enable alpha individuals to regulate population breeding rates (e.g., wolves and lions), is misleading in the context of domestic cat clowders that tolerate co-inhabitants without limiting recruitment rates. Population reduction models suggest that sterilization is typically less effective than culling [26,27], particularly in polygamous species where even high sterilization rates of males barely reduce population fecundity [28].

### 3.3. Food is Limiting

Claims that neutered cats defend food resources and limit the immigration of cats into TNR-managed areas are disproven by cat populations responding positively to increased food resources [2]. If cats were territorial, even colonial, dense clowders would be prevented from forming at sites with subsidised food by aggression from incumbent dominant cats. However, high cat densities at sites with food subsidies, including many reviewed in the TNR literature, demonstrate that even intact cats do not defend food resources. Neutered cats with lowered social status or less reason to maximise their reproductive success are even less likely to defend mates or food from interlopers, thus negating one of the main purported mechanisms of limiting cat clowder size through TNR. Aggregations of domestic cats only occur where supplementary (non-hunted) food is available and the size of these clowders [29] and domestic cat recruitment is proportional to food availability, as demonstrated by dramatic increases in cat populations when their prey are abundant [30,31]. Hence, arguably the largest, yet often overlooked, problem with TNR is the deliberate provisioning of food resources to released cats. Furthermore, untrapped, dumped, or immigrant intact cats will increase their breeding potential through access to supplementary food. Removal of these food subsidies will dramatically reduce recruitment rates and the number of unowned cats.

## 4. Environmental Flaws Associated with TNR

TNR is entirely focused on cats and not on other species or the environment in which cats live. Hence, TNR does not address predation of wildlife, disease transmission, or the environmental impact of sourcing food for stray cats or their faecal accumulation and runoff.

### 4.1. Incidence of Disease

A range of diseases that cause both acute and chronic health impacts to humans is demonstrably higher in aggregations of outdoor cats that use communal latrines [32] than in household pets, and cats that have access to the outdoors have markedly higher parasite loads than pets kept indoors [33].

### 4.2. Wildlife Impacts

There is no reason or data to disprove the assertion that neutered cats released to the environment would exert any lower predation pressure on wildlife than other cats [34,35,36]. For example, a neutered cat was responsible for the predation and abandonment of a colony of threatened fairy terns [37]. Even well fed pet cats hunt [38,39], and despite per capita kill rates of pet cats being only one quarter those of feral cats, the much higher densities of pet and stray cats account for wildlife predation rates in urban areas that are more than 28 times greater than for feral cats in natural environments [40].

### 4.3. Marine Resource Depletion and Food Waste

Production of cat food may have significant impacts on fisheries and fish conservation [41,42], which is an impact far more tenuous to justify for feral or stray cats than if these resources were depleted for household pets. Surplus food not eaten by stray cats is also wasted or potentially increases populations of scavenging species.

## 5. Ethical and Social Flaws Associated with TNR

### 5.1. “Dumping”

Dumping of unwanted cats at recognised TNR clowder sites represents an outsourcing of responsibility by irresponsible cat owners and promulgates the stray cat problem. This irresponsible dumping of unwanted cats has been identified as a cause of the failure of several well documented TNR campaigns [16,43].

### 5.2. Emotional and Financial Stress

The “no-kill” ideology that underpins the promotion of TNR preys on well-intentioned and caring vulnerable members of society who sometimes spend scarce funds on cats to their own detriment.

### 5.3. Increased Suffering

Net pain and suffering are higher where unowned cat populations are maintained or increased by feeding. Reduction in overall animal pain and suffering are key reasons for informed cat welfare organisations including PeTA and the Australian Veterinary Association opposing TNR [44].

### 5.4. Conflict of Interest

Peter Wolf, as an advocate for TNR, has an apparent conflict of interest because he is being paid by Best Friends, a lobbying group that strongly supports the no-kill ideology [45]. In our opinion, lobbyists should be free to discuss policy and management, but authors misusing science for lobbying and not declaring their real or perceived conflict of interest in scientific papers is unethical.

## 6. Economic Flaws in TNR

Proponents of TNR often claim that the approach is economically viable. However, TNR is neither low cost by itself nor in comparison to other methods. First, TNR is labour intensive, requiring time from a veterinarian or licensed professional to anesthetize and surgically alter each cat. While vets may volunteer their time, in full economic accounting, this time must be valued. Moreover, many TNR clinics are still funded or subsidized to pay for medicine and supplies as well as veterinarians’ time; in some cases, cat food is also donated including by pet food wholesalers and retailers who have a vested interest in maintaining high cat populations. Second, no TNR advocacy groups have published peer reviewed economic analyses demonstrating that their approach makes fiscal sense. When coupled with the lack of success in reducing stray cat numbers, TNR is shown to be an inefficient use of capital [20].

## 7. The BCC Case Study

The Brisbane City Council (BCC) is responsible for ensuring that individuals or organisations uphold their general biosecurity obligations under the Queensland Biosecurity Act 2014. Following complaints from concerned Brisbane residents and business owners, BCC initiated proactive management of “non-domestic cats” (renamed strays for consistency with this paper) in the Coopers Plains district in 2013, which then expanded to other council districts. Prior to the commencement of trapping programs, residents within 250 m of a trap site are sent a letter notifying them that BCC will be trapping and providing education on the resident’s responsibilities to keep their domestic pets on their property. For a cat, this means it must be inside the house or in an adequate outdoor enclosure.

All trapped cats are transported to a BCC facility, scanned for microchips (compulsory in Queensland for all cats prior to 12 weeks of age in accordance with the Animal Management (Cats and Dogs) Act 2008), and assessed for other ownership indicators such as collars or desexing tattoos. The assessing officer takes into account the previous history of the area, phone calls received from concerned residents when they receive a notification letter, and discussions with local residents and workers. Cats with any evidence of ownership are transferred to BCC’s animal rehoming centre in order to reunite the cat with its owner. To reclaim their pet, owners are required to pay a release fee and potentially, a penalty, and also sign a document stating that they will comply with the requirements under BCC local laws, which includes that they maintain an adequate enclosure for their cat.

Most (391 of 401) of the stray cats captured at Coopers Plains were euthanased as soon as possible, without a holding period, in order to reduce stress. The success of this program is evidenced by a reduction in public complaints about stray cats and captures after 2016 (Figure 1) along with anecdotal increases in sightings of wildlife including bush stone curlews (D. Franks, pers. comm.).

The wider BCC program trapped 4685 stray and 700 domestic cats from 50,887 trap nights from 2014 to 2019 at a cost of AUS $1.3 M, including considerable resources for public education. An example of the efficacy of the BCC’s proactive trapping program was the complete removal of a clowder of 79 stray cats from Richlands within 10 weeks of its discovery in April 2020. In addition to removing stray cats, the BCC program has led to the improved management of domestic cats. Public education has been vital in changing behaviour and has proven to be sufficient (in most cases) for owners to comply with the local law. However, despite these efforts, some strays were repeatedly trapped or recorded as a result of the persistent feeding and release of non-domestic cats. To address community concern and prevent further feeding and release of strays, BCC carried out a program of compliance and enforcement action to improve their legislated biosecurity risk management. Enforcement has been critical for reducing stray cat populations, particularly through reducing food subsidies, inhibiting cats’ ability to reproduce at high rates and allowing for higher success rates when trapping (D. Franks, pers. comm.). Since 2017, 52 convictions totalling $27,000 and a three-month imprisonment term have been handed down, highlighting the court’s view on the seriousness of these offences.

Together, the local cat management laws, awareness, prosecutions and targeted control of unowned cats have been integral to the observed behavioural change and reduced non-compliance, and complaints to the BCC. As of July 2020, there were no known clowders of cats remaining in Brisbane (Bill Manners, pers. comm.).

## 8. The Way Forward

We offer the following hierarchical strategy for effectively and ethically meeting the mutually agreed objective of sustainably reducing the number of unowned urban and suburban cats.

### 8.1. Eliminate Deliberate Feeding and Reduce Inadvertent Feeding of Unowned Cats

Proactive management of food scraps through exclusion fencing of dumps, diligent management of anthropogenic food wastes, and prohibition of feeding of uncontained cats is likely the single most important contributor to reducing the number of unowned urban cats. To be effective, laws or rules prohibiting feeding of unowned cats need to be reinforced by prosecution, as per the Brisbane City Council experience.

### 8.2. Compulsory Registration and Desexing

All pet cats should be identifiably registered, like dogs [24], thus, the onus shifts from cat managers to cat owners to confirm ownership. Only registered breeders or cat refuges should be allowed to sell cats and giving away cats should be illegal. All cats should be desexed, registered and vaccinated before sale. Registration, which is being increasingly adopted by a number of Australian local governments, would dramatically reduce holding times, expense and stress to cat managers/welfare practitioners, and cats. The registration of cats also helps to shift the mindset of those cat owners who consider cats to be a low effort and low-cost pet option. Improving the status and management of pet cats will help reduce leakage to the stray and feral populations.

### 8.3. Containment

Cats should be contained in yards or houses (again, just like dogs), which is advantageous for cat welfare, cat owners’ experience, logistics of managing unowned cats and minimising human health and wildlife death and risk [5,33,46,47]. Containment of pet cats is already endorsed by most cat advocates and an increasing number of Australian local governments [48], and it drastically simplifies the management of unowned cats.

### 8.4. Enhanced Adoption of Kittens

Adoption of desexed and vaccinated stray cats is the ideal way to connect willing owners with needy pets and contribute to a reduction in the number of stray cats. Pet shops and cat welfare organisations should maintain a register of both cats for adoption and potential owners looking to adopt different categories of cats in the local region, recognising that unowned kittens are typically easier to habituate to domesticity and indoor life than older unowned cats.

### 8.5. Ethical and Efficient Control of Surplus Cats

The RSPCA-endorsed Humaneness index identifies the duration of capture as a significant determinant of welfare outcomes for trapped animals [49], which is consistent with the rapid assessment and management of stray cats instead of protracted retention and veterinary treatment. In line with the successful BCC model, an unowned cat from a neighbourhood where cats currently held in adoption outnumber realistic demand (typical adoption rates/expressions of interest/waiting lists) should be immediately euthanased, thus reducing the ongoing stress to cats and carers and costs to municipal agencies and cat welfare groups.

### 8.6. Optimum Monitoring

Where diligent public reporting and proactive and prompt unowned cat removal is achievable, such as practiced by the BCC, resourcing and managing privacy issues surrounding camera monitoring of clowders are unnecessary and inefficient (B. Manners, pers comm.). However, at sites where stray cats are not immediately removed, management effectiveness should be monitored directly according to scientific principles, rather than using indirect and potentially misleading indicators such as euthanasia rates from reactive trapping.

These six initiatives provide a pragmatic and informed framework for stray cat management in Australia. Collectively, they are predicted to significantly reduce suffering and the range of animal welfare, disease, conservation, stress and social issues associated with stray cats.

## Figures and Tables

**Figure 1 animals-10-01525-f001:**
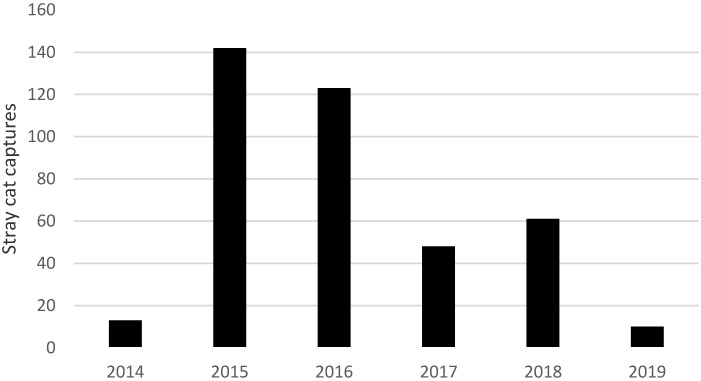
Stray cat captures by Brisbane City Council from Coopers Plains district 2014–2019.

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
