# Peer review of "Reply to Wolf et al.: Why Trap-Neuter-Return (TNR) Is Not an Ethical Solution for Stray Cat Management"

_animals, 2020, doi:10.3390/ani10091525_

Round 1

Reviewer 1 Report

animals-910085 Read et al.

Debate over claims in published papers is an important part of the development of scientific ideas, including the testing of evidence and argument. In this context, the manuscript advances discussion of the suitability of TNR for managing populations of free-roaming, unowned cats in Australia in particular and also globally. I doubt that it has the last word on many points, but it is of considerable value in clarifying arguments and positons.

The authors may wish to consider these points in revising their paper:

Simple summary

  • I can’t recall seeing such a succinct, effective simple summary.

Abstract

  • The use of ‘re-abandonment’ in line 22 may well be the authors’ opinion of the practice of TNR, but it is emotive and provocative for the abstract where the evidence to support the view cannot be presented. Perhaps use ‘return’ here and raise the idea of re-abandonment later when evidence for the position can be presented.

Introduction

  • Line 33 – I don’t think you really mean ‘minimising animal welfare.’ Don’t you actually mean that animal welfare is improved or optimised? Or you could say that wish to minimise animal welfare concerns.

Critique of Wolf et al.

  • Line 65 – Is ‘sustainable reduction’ the right term here? To me, it implies a rate of reduction maintained over time, rather than reducing to a low level and maintaining that low level.
  • Lines 75-86 – There are several contentious claims here that should be explicated, with consideration of relevant literature. Papers, including reviews, cited to support population reductions caused by TNR or its theoretical efficacy include: Kreisler et al. 2019, Boone et al. 2019, Hamilton 2019, Wolf and Hamilton 2020. For empirical data, Spehar and Wolf (2018) claim elimination of eight colonies and refer to two other studies (refs 19 and 20 in their paper, I did not look them up) that claimed elimination. These studies need to be refuted if the reference to recent reviews includes all reviews, not just those against TNR, and if the claim of no eliminations within the lifetime of a cat (which could be up to 20 years) is to be supported.
  • Lines 101-103 – I thought from Calver et al. (2020) that euthanasia rates were now well under 60% in RSPCA shelters across Australia.
  • Lines 121-126 – Why is the difficulty in trapping cats not a problem for the approach you advocate? Won’t some cats be hard to trap and remove?
  • Lines 134 -137 – Check the modelling in Boone et al. (2019), which I think argues for an advantage for TNR over culling.
  • Lines 161-166 – There is a compelling case from Western Australia that supports your contention for hunting by desexed cats. A desexed stray cat was instrumental in causing the total failure of a breeding colony of Fairy Terns (Greenwell et al. 2019). Or you could look to any of the studies of anti-predator devices in Australia, where the animals volunteered for the studies were all desexed but also formidable hunters (e.g. Hall et al. 2015, Calver et al. 2007).
  • Lines 181-184 – How do you know that welfare is depressed in TNR colonies? Kreisler et al. (2019) claim improved welfare, as do Gilhofer et al. (2019), Gunther et al. (2018) and Finkler et al. (2010). Against this you could refer to Castro-Prieto and Andrade-Nunez (2018) for examples of poor health in TNR cats, or Crawford et al. (2020) for details of health risks and shortened lifespans for stray cats in Australia.
  • Lines 185-188 – This is pointed and personal. You’ve singled out Wolf from all the authors of Wolf et al. (2019) as having a conflict of interest and being a non-scientist, whereas Spehar is ‘an independent researcher,’ Norris is from the RSPCA, Swarbrick appears to be an optometrist, while Rand claims both a welfare and an academic address. This looks like a vendetta against Wolf. If you want to take this dangerous path (and bear in mind the possibility of a defamation case in Australia, which is far easier to pin than it is in the US), then: (i) make clear that you are concerned with people not acknowledging that they are salaried employees of advocacy groups as a conflict of interest (and would not this also implicate Rand and Norris?), (ii) be very sure of yourself regarding branding Wolf a ‘non-scientist,’ (do you know his CV? And what is a scientist anyway?), (iii) state that this is your opinion (from a legal point of view, you are entitled to an opinion, but beware making a statement of fact that you may be compelled to prove). Finn (2019) gives a good coverage of the potential pitfalls.
  • Lines 188-199 – The whole of section 5 of Wolf et al. (2019) is concerned with economic arguments, so I was surprised that you do not engage with them directly.
  • Where you use personal communications, it would be good to give an email address or other means of contact with these people. The purpose of a reference is so that readers can trace sources. This can’t be done easily from just a name and an affiliation.

References

  • Check that the Latin names in the references are italicised.

References used in this review

Boone JD, Miller PS, Briggs JR, Benka VAW, Lawler DF, Slater M, Levy JK, Zawistowski S. 2019. A Long-Term Lens: Cumulative Impacts of Free-Roaming Cat Management Strategy and Intensity on Preventable Cat Mortalities. Frontiers in Veterinary Science 6.

Calver M, Thomas S, Bradley S, McCutcheon H. 2007. Reducing the rate of predation on wildlife by pet cats: The efficacy and practicability of collar-mounted pounce protectors. Biological Conservation 137:341-348.

Calver MC, Crawford HM, Fleming PA. 2020. Response to wolf et al.: Furthering debate over the suitability of trap-neuter-return for stray cat management. Animals 10.

Castro-Prieto J, Andrade-Núñez MJ. 2018. Health and ecological aspects of stray cats in old san juan, puerto rico: Baseline information to develop an effective control program. Puerto Rico Health Sciences Journal 37:110-114.

Crawford HM, Calver MC, Fleming PA. 2020. Subsidised by junk foods: factors influencing body condition in stray cats (Felis catus). Journal of Urban Ecology:1-17.

Finkler H, Terkel J. 2010. Cortisol levels and aggression in neutered and intact free-roaming female cats living in urban social groups. Physiology and Behavior 99:343-347.

Finn HC. 2019. The defamatory potential of ad hominem criticism: Guidance for advocacy in public forums. Pacific Conservation Biology 25:92-104.

Gilhofer EM, Windschnurer I, Troxler J, Heizmann V. 2019. Welfare of feral cats and potential influencing factors. Journal of Veterinary Behavior 30:114-123.

Greenwell CN, Calver MC, Loneragan NR. 2019. Cat gets its tern: A case study of predation on a threatened coastal seabird. Animals 9.

Gunther I, Raz T, Klement E. 2018. Association of neutering with health and welfare of urban free-roaming cat population in Israel, during 2012-2014. Preventive Veterinary Medicine 157:26-33.

Hall CM, Fontaine JB, Bryant KA, Calver MC. 2015. Assessing the effectiveness of the Birdsbesafe® anti-predation collar cover in reducing predation on wildlife by pet cats in Western Australia. Applied Animal Behaviour Science 173:40-51.

Hamilton F. 2019. Implementing Nonlethal Solutions for Free-Roaming Cat Management in a County in the Southeastern United States. Frontiers in Veterinary Science 6.

Kreisler RE, Cornell HN, Levy JK. 2019. Decrease in population and increase in welfare of community cats in a twenty-three year trap-neuter-return program in Key Largo, FL: The ORCAT program. Frontiers in Veterinary Science 6.

Spehar DD, Wolf PJ. 2018. A case study in citizen science: The effectiveness of a trap-neuter-return program in a Chicago neighborhood. Animals 8.

Wolf PJ, Rand J, Swarbrick H, Spehar DD, Norris J. 2019. Reply to crawford et al.: Why trap-neuter-return (TNR) is an ethical solution for stray cat management. Animals 9.

Wolf PJ, Hamilton F. 2020. Managing free-roaming cats in U.S. cities: An object lesson in public policy and citizen action. Journal of Urban Affairs.

Author Response

Debate over claims in published papers is an important part of the development of scientific ideas, including the testing of evidence and argument. In this context, the manuscript advances discussion of the suitability of TNR for managing populations of free-roaming, unowned cats in Australia in particular and also globally. I doubt that it has the last word on many points, but it is of considerable value in clarifying arguments and positions.

The authors may wish to consider these points in revising their paper:

Simple summary

  • I can’t recall seeing such a succinct, effective simple summary.

Abstract

Point 1 The use of ‘re-abandonment’ in line 22 may well be the authors’ opinion of the practice of TNR, but it is emotive and provocative for the abstract where the evidence to support the view cannot be presented. Perhaps use ‘return’ here and raise the idea of re-abandonment later when evidence for the position can be presented.

 Response     Re-abandonment has been used before to accurately describe TNR but in the interests of retaining an unemotive appraisal in our rebuttal of the Wolf et al manuscript we have revised, as suggested, to ‘returning cats to stray populations’.

Introduction

  • Point 2 Line 33 – I don’t think you really mean ‘minimising animal welfare.’ Don’t you actually mean that animal welfare is improved or optimised? Or you could say that wish to minimise animal welfare concerns.
  • Response Good point, revised as suggested

Critique of Wolf et al.

  • Point 3 Line 65 – Is ‘sustainable reduction’ the right term here? To me, it implies a rate of reduction maintained over time, rather than reducing to a low level and maintaining that low level.
  • Response We were following the terminology of Wolf et al with ‘sustainable reduction’ but agree that the intent of stray cat management should be better worded to “reduce stray cats numbers to very low levels that are sustained indefinitely”,   and have revised accordingly.

  • Point 4 75-86 – There are several contentious claims here that should be explicated, with consideration of relevant literature. Papers, including reviews, cited to support population reductions caused by TNR or its theoretical efficacy include: Kreisler et al. 2019, Boone et al. 2019, Hamilton 2019, Wolf and Hamilton 2020. For empirical data, Spehar and Wolf (2018) claim elimination of eight colonies and refer to two other studies (refs 19 and 20 in their paper, I did not look them up) that claimed elimination. These studies need to be refuted if the reference to recent reviews includes all reviews, not just those against TNR, and if the claim of no eliminations within the lifetime of a cat (which could be up to 20 years) is to be supported.

Response   We have weighed up the options of citing and rebutting claims in each of these papers but have decided in the interests of brevity in this rebuttal to retain our defendable statement that “the weight of scientific evidence” shows that TNR is not effective

  • Point 5 Lines 101-103 – I thought from Calver et al. (2020) that euthanasia rates were now well under 60% in RSPCA shelters across Australia.

Response  60% was the figure cited by Wolf et al. but the actual percentage is not relevant to our argument : revised to “which is  not assessable by the percentage of cats impounded by local government that are euthanized.”

  • Point 6 Lines 121-126 – Why is the difficulty in trapping cats not a problem for the approach you advocate? Won’t some cats be hard to trap and remove?
  • Response We have added the following sentence to explain: “Lethal control, which could include techniques that either do not require cats to be captured or at most require one capture, do not face the same limitations as TNR, which requires multiple captures to retain effective immunisations.”
  • Point 7 Lines 134 -137 – Check the modelling in Boone et al. (2019), which I think argues for an advantage for TNR over culling.
  • Response Boone et al 2019 modelled the effect of different management actions (including low and high (75%) sterilization rates of TNR) on preventable cat deaths, rather than populations of stray cats. We agree with Wolf et al. and Campbell et al. that stray cat population size should be the key metric of success of stray cat management. Boone et al. concede “The choice of management strategy should ideally incorporate multiple factors, including population outcome, cat welfare, cat impacts on wildlife, cost effectiveness, ethics, practicality, tractability, likelihood of success, and political/public support”. We agree that these other factors are integral to determining optimal stray cat management and don’t consider that Boone’s modelling contradicts our position that TNR programs (not hypothetical modelling) have not been demonstrated to provide sustainable and economically viable management of stray cat populations (with few exceptions where rehoming was more influential than achievable neutering rates) 
  • Point 8 Lines 161-166 – There is a compelling case from Western Australia that supports your contention for hunting by desexed cats. A desexed stray cat was instrumental in causing the total failure of a breeding colony of Fairy Terns (Greenwell et al. 2019). Or you could look to any of the studies of anti-predator devices in Australia, where the animals volunteered for the studies were all desexed but also formidable hunters (e.g. Hall et al. 2015, Calver et al. 2007).
  • Response Good point, we have now cited the Greenwell paper “For example a neutered cat is believed responsible for predation and abandonment of a colony of threatened fairy terns [36A]
  • Point 9 Lines 181-184 – How do you know that welfare is depressed in TNR colonies? Kreisler et al. (2019) claim improved welfare, as do Gilhofer et al. (2019), Gunther et al. (2018) and Finkler et al. (2010). Against this you could refer to Castro-Prieto and Andrade-Nunez (2018) for examples of poor health in TNR cats, or Crawford et al. (2020) for details of health risks and shortened lifespans for stray cats in Australia.

Response  Welfare is compromised in TNR clowders. Crawford et al, 2020 noted an increase in GI parasites and to quote the article ‘alarmingly, 57.5% of strays were scavenging vast amounts of refuse, including life-threatening items in volumes that blocked their gastrointestinal tracts. Castro-Prieto and Andrade-Nunez (2018) noted 70% of the cats that were captured were neutered, and

21% of these individuals exhibited very poor physical condition, including skin problems,

scars, underweight, and blindness.

  • Point 10 Lines 185-188 – This is pointed and personal. You’ve singled out Wolf from all the authors of Wolf et al. (2019) as having a conflict of interest and being a non-scientist, whereas Spehar is ‘an independent researcher,’ Norris is from the RSPCA, Swarbrick appears to be an optometrist, while Rand claims both a welfare and an academic address. This looks like a vendetta against Wolf. If you want to take this dangerous path (and bear in mind the possibility of a defamation case in Australia, which is far easier to pin than it is in the US), then: (i) make clear that you are concerned with people not acknowledging that they are salaried employees of advocacy groups as a conflict of interest (and would not this also implicate Rand and Norris?), (ii) be very sure of yourself regarding branding Wolf a ‘non-scientist,’ (do you know his CV? And what is a scientist anyway?), (iii) state that this is your opinion (from a legal point of view, you are entitled to an opinion, but beware making a statement of fact that you may be compelled to prove). Finn (2019) gives a good coverage of the potential pitfalls.
  • Response Peter Wolf has been ‘singled out’ because he is the lead author. We all feel that PW is free to advocate his ideologies through his unregulated and non-peer reviewed Voxfelina website, which is clearly advocacy-based and focused on tearing down academic research that does not support his views. However, it is unethical to extend his unsubstantiated and non-scientific rebuttals to scientific journal articles, including the Campbell et al paper, at least without declaring a conflict of interest.  We know less about the motivations, science background or role in preparing the manuscript of the coauthors but the ethical points we are making are consistent. Scientific journals should be used for transparently presenting and discussing facts. Funding sources or employment that could compromise this science should be reported as a potential conflict of interest – We have revised slightly as follows:
  • Peter Wolf, as an advocate for TNR, has an apparent conflict of interest through being paid by Best Friends, a lobbying group that strongly supports the ‘no kill’ ideology [44]. In our opinion, lobbyists should be free to discuss policy and management, but authors misusing science for lobbying and not declaring their real or perceived conflict of interest in scientific papers is unethical”.
  • Point 11 Lines 188-199 – The whole of section 5 of Wolf et al. (2019) is concerned with economic arguments, so I was surprised that you do not engage with them directly.

Response  We have now addressed this point directly with the addition of the following Statement and Response:

  5S        Wolf et al. state “The proposed ‘targeted adoption’ program, which involves “rehabilitation” of unsocialized cats, would very likely result in much greater costs (as well as overcrowded shelters and an increase in the number of cats killed because of space limitations). The authors acknowledge the “clear need for economic research on the relative costs and effectiveness of different proposed strategies for reducing numbers of stray cats in Australian cities” even as they appear to accept, uncritically, the results of a dubious model based on a “single super colony [of 30,000 cats]” [29]. We too welcome additional economic analysis; however, the available evidence strongly suggests that the costs associated with expanded adoption efforts will likely exceed the cost of TNR.”

5R          We appreciate that Wolf et al. agree on the need for further economic research. However, we would note that the authors provide no data or literature to support their claim that the targeted adoption program or expanded adoption programs would cost more that TNR. While an adoption program is not free, full cost accounting in a comparative manner needs to be done in order to demonstrate what approaches truly cost. We would further note that the ‘dubious’ study [29] did show full cost accounting and was published in a well regarded peer reviewed journal. Finally, simply providing accounting for costs is incomplete as the economic analyses need to be contextualized with respect to what they achieve in terms of reducing the number of stray cats. For instance, an adoption program may have a greater total cost, but be less expensive on a per capita basis, and therefore more efficient.

  •  
  • Point 12 Where you use personal communications, it would be good to give an email address or other means of contact with these people. The purpose of a reference is so that readers can trace sources. This can’t be done easily from just a name and an affiliation.
  • Response Bill Manners and Dan Fanks from BCC have been approached to see if they are happy for their email addresses to be published

References

Check that the Latin names in the references are italicised